# Temporal Aggregation Effects in Spatiotemporal Traffic Modelling

**DOI:** 10.3390/s20236931

**Published:** 2020-12-04

**Authors:** Dmitry Pavlyuk

**Affiliations:** Transport and Telecommunication Institute, LV-1019 Riga, Latvia; Dmitry.Pavlyuk@tsi.lv

**Keywords:** spatiotemporal models, temporal aggregation, forecasting accuracy, big data, urban traffic modelling

## Abstract

Spatiotemporal models are a popular tool for urban traffic forecasting, and their correct specification is a challenging task. Temporal aggregation of traffic sensor data series is a critical component of model specification, which determines the spatial structure and affects models’ forecasting accuracy. Through extensive experiments with real-world data, we investigated the effects of the selected temporal aggregation level for forecasting performance of different spatiotemporal model specifications. A set of analysed models include travel-time-based and correlation-based spatially restricted vector autoregressive models, compared to classical univariate and multivariate time series models. Research experiments are executed in several dimensions: temporal aggregation levels, forecasting horizons (one-step and multi-step forecasts), spatial complexity (sequential and complex spatial structures), the spatial restriction approach (unrestricted, travel-time-based and correlation-based), and series transformation (original and detrended traffic volumes). The obtained results demonstrate the crucial role of the temporal aggregation level for identification of the spatiotemporal traffic flow structure and selection of the best model specification. We conclude that the common research practice of an arbitrary selection of the temporal aggregation level could lead to incorrect conclusions on optimal model specification. Thus, we recommend extending the traffic forecasting methodology by validation of existing and newly developed model specifications for different temporal aggregation levels. Additionally, we provide empirical results on the selection of the optimal temporal aggregation level for the discussed spatiotemporal models for different forecasting horizons.

## 1. Introduction

Traffic flow forecasting is a classic problem of transportation engineering. Recent developments in intelligent urban transportation systems led to the availability of an overwhelming amount of geospatial data, collected in modern cities by a wide network of distributed traffic sensors and tracking devices [1]. Thus, the mainstream of academic research into traffic flows shifted from univariate modelling of spatially disconnected road segments to spatiotemporal analysis, which utilizes potential dependencies between different locations [2].

Spatiotemporal traffic analysis requires careful identification of the spatial structure—a set of dependencies between traffic characteristics at neighbour or remote road segments [3]. Although modern spatiotemporal models provide various treatments for the incorporation of spatial information, the problem is still emerging for dynamic spatial environments. Since Okutani and Stephanedes [4] directed attention to spatial dependencies between traffic flows and practical utility of this information for traffic forecasting, many researchers utilized information about the spatial structure in their methodologies. Applied methodologies could be tentatively classified into two approaches, as is classical for spatial econometrics—specific-to-general and general-to-specific.

The specific-to-general approach is based on the gradual extension of univariate traffic flow models with carefully selected spatial dependencies (e.g., dependencies between the target and upstream road segments). Stathopoulos and Karlaftis [5] and Vlahogianni et al. [6] successfully utilized a spatial dependency between the target road segment and its immediate upstream neighbour for improvement of the model forecasting accuracy (time series models and neural networks respectively). This type of spatial dependency directly follows from the kinematic traffic flow theory and its discovered utility is natural. Further studies include downstream [7] or bi-directional [8] spatial dependencies between neighbour road segments. Downstream dependencies are also well theoretically supported by the back propagation of traffic congestions. The next methodological step in this direction is application of spatial econometric advances. A set of spatial dependencies is generalized in a form of a matrix of spatial weights, which contains inverse spatial dependencies between every pair observed locations. Usually, this matrix is constructed exogenously and based on geographical distances or travel times between the locations. The space–time autoregressive model (STAR), introduced by Cliff et al. [9], and its direct generalization to the space–time autoregressive integrated moving average model (STARIMA) [10], are based on exogenously provided spatial weight matrices and widely used for traffic flow forecasting [11,12] (see Elhorst [13] for a comprehensive discussion of STAR-based models and their applications). One of the main challenges for STAR-based models is a proper specification of the spatial weight matrix: spatial dependencies are naturally unstable over time in a dynamic traffic flow environment, and so spatial weights should be considered as dynamic. This issue can be solved by the application of threshold or regime-switching models [14] or models with dynamic speed-dependent spatial weights [12]. 

Studies that utilize the second, general-to-specific, approach start from the most general model, where spatial dependencies could potentially exist between any road segments, and gradually reduce the number of spatial links. The presence of spatial relationships between remote road segments is usually explained [2] by common traffic patterns (for example, event-specific traffic flows to the city centre from outskirts) or by the complementary nature of road segments (thus negative spatial links are also possible). The most widely used statistical implementation of this approach is the popular vector autoregressive (VAR) model [15] and its generalization to the vector autoregressive moving average (VARMA) model [16]. Several recent studies utilize the VAR model and its modifications for identification of spatial relationships in the traffic flow [11,14,17,18,19,20,21]. Despite the obvious advantages of this data-driven approach, highly dimensional VAR models suffer from an extremely large number of parameters and the overfitting problem. Overfitting is especially critical for spatial traffic flow modelling, where data series come from thousands of spatial locations (sensors).

One of emerging approaches to overcome overfitting problems of highly dimensional VAR models is limiting the number of possible dependencies using additional exogenous information or statistical techniques (regularization). VAR models, where the number of parameters is significantly limited by introduced restrictions, are called sparse. Clements and Hendry [22] showed that the forecast accuracy can be improved by imposing zero restrictions on coefficients that are close to zero even if the true data generating process has none of these restrictions. Statistical regularization (such as lasso, least absolute shrinkage and selection operator, or the elastic net) is the most widely used technique for increasing the sparsity of VAR models (for example, Kamarianakis et al. [14] applied lasso regularization to discover significant spatial links between road segments and to increase model forecasting performance). Statistical regularization can be implemented in two steps, where the first step is a correlation-based identification of spatial dependencies and the second one is the estimation of the sparse VAR model with the most significant features and lags (see [23] for a possible implementation of this approach). Recent studies [21,24,25] provided empirical evidence of the successful utilization of correlation-based sparsity for traffic flow modelling. Although statistical regularization was successfully applied in practice, direct introduction of a spatial structure looks like a more information-thrifty approach. Spatially restricted VAR models introduce zero restrictions on coefficients on the basis of an exogenously provided spatial structure. Spatial restrictions can be implemented using the same principles as spatial weights—using the distance- or travel-time-based road network topology [19,20]. Note that spatially restricted VAR models can be considered as a “softer” and more flexible way to introduce spatial dependencies—in contrast to classical spatial econometric models, where the spatial structure should be strictly defined via spatial weights, spatially restricted models define only spatial links, where a dependency could be (but is not necessary to be) discovered.

The spatiotemporal structure also plays an important role in modern deep learning models of traffic flows [26,27]. Recently developed deep learning model specifications [28,29,30,31] utilize the graph convolution technique: the models are trained for a spatiotemporal graph of relationships that appear between road segments with different temporal lags. Although identification of the most important relationships can be implemented internally (for example, using the spatiotemporal attention mechanism [32]), these advanced models also require external definition of the spatiotemporal structure.

Regardless of the learning approach chosen, the resulting spatial structure of traffic flows is complex and includes both static and dynamic components—it strongly depends on the road network configuration and current traffic conditions. In addition, the spatial structure highly depends on the temporal aggregation level (for example, sets of spatially linked locations are completely different for 1 min and for 5 min time intervals). Within the scope of this research, we stress that the structure of spatial dependencies strongly depends on a selected temporal aggregation level. For example, if we consider travel times between road segments as a metric for spatial weights’ construction, first-order neighbour segments will be completely different for 1 and 5 min temporal aggregation levels. Thus, the problem of temporal aggregation plays a critical role for introducing the spatial structure into traffic flow models. The importance of temporal resolution for traffic flow modelling is repeatedly accentuated by Vlahogianni et al. [33,34,35]. Analysis of temporal aggregation effects has a long history in the literature on forecasting methodology, but is still emerging [36]. Early [37,38,39,40] and recent [41,42] methodological studies provided treatments for temporal aggregation of univariate time series models. Temporal aggregation of multivariate models is also well covered in the theoretical literature [43,44,45]; in particular, Breitung and Swanson [44] provided theoretical evidences for a dependence between Granger causal relationships and temporal aggregation of multivariate time series. These studies generally advise using disaggregated data for model estimation to prevent losses of information and increase forecasting performance. Although advantages of temporal aggregation have no theoretical support, several recent studies [46,47,48,49,50] provide empirical evidence of the better forecasting performance of models, estimated on the basis of aggregated data. Kourentzes [48,51] explained these empirical findings by easier identification of different time series components using different temporal aggregation levels (i.e., trends could be more easily identified for aggregated data, where random disturbances have lower impact). This argument is important in the context of this research, because we expect that the structure of spatial dependencies has positive forecasting accuracy effects for specific aggregation levels only. 

Although the importance of temporal aggregation was highlighted by several recent studies in the traffic forecasting domain (e.g., Vlahogianni and Karlaftis [35]), many empirical studies that utilise spatiotemporal models are focused on a specific temporal aggregation level only. Suggesting a new methodological approach, the authors arbitrarily select a temporal aggregation level for their experimental part (i.e., 5 min time intervals for short-term forecasting) and make potentially improper generalisations of obtaining results.

Summarising the literature review, we conclude that:The spatiotemporal structure of traffic flow dependencies plays an important role in the modern traffic forecasting methodology.Definition of the spatiotemporal structure is a complex problem, whose solution depends on the selected level of temporal aggregation of data series, among other parameters.The optimal level of temporal aggregation is subject to many problem- and case-specific factors and should be estimated empirically.Despite the acknowledged importance of temporal aggregation, its role is rarely addressed in methodological literature on traffic flow forecasting.

In this research, we provide empirical evidence of the importance of temporal aggregation for estimating a forecasting accuracy of spatiotemporal models. We argue that the analysis of spatiotemporal model specifications, conducted for an arbitrary specified temporal aggregation level, can easily lead to misleading conclusions. Thus, the main research question addressed in this study is:Does the temporal aggregation level play a crucial role for estimating forecasting accuracy of spatiotemporal traffic flow models? If yes—is it possible to empirically estimate the optimal temporal aggregation level, given the model specification and required forecasting horizon?

To the best of our knowledge, there are no previous studies focusing on the importance of temporal aggregation for spatiotemporal traffic forecasting models. The fact of this importance is acknowledged in recent literature, e.g., Vlahogianni and Karlaftis [35] stated that “further research is needed to determine the optimum aggregation level with respect to different transportation modelling applications”. At the same time, the majority of recent methodological studies did not include the level of temporal aggregation into a set of tuned parameters (usually, this set is limited by different forecasting horizons, definitions of the spatiotemporal structure, and model-specific hyperparameters). We state that this exclusion can lead to incorrect conclusions on models’ forecasting performance and inappropriate model selection.

## 2. Materials and Methods

This section contains a detailed description of the research dataset and methodology, including aspects of data pre-processing, definition of spatial structures, model specifications, and measurement of the forecasting accuracy.

### 2.1. Research Data

The dataset was collected by sensors deployed in the Minneapolis urban area and managed by the Minnesota Department of Transportation (MnDOT). The sensors are regular traffic detectors that include inductive loops installed in the pavement and constantly test their inductance. The inductance is raised when a massive metal object passes the loop (the sensors are tuned to identify scooters and larger vehicles), and this signal is used for traffic measurement. Sensors collect information on the traffic volume (number of cars passed the loop) and the occupancy (percentage of time when a vehicle is located in the loop). Further, these raw indicators can be used for traffic classification, estimation of traffic speed, and other traffic values. This study was purely based on raw traffic counts, reported by the sensors and aggregated to 1 min time frames. These data are publicly available and were recently utilized in several studies [21,24,52]. We utilized traffic volume data for 5 weeks (35 days) from 26 February 2017 to 1 April 2017. Data were collected from the MnDOT system with 1 min temporal aggregation, and further aggregated for 2, 3, 4, 6, and 12 min intervals.

We analysed two road network segments to discover the effects of temporal aggregation in different spatial settings. The first road network segment represents the sequential spatial structure and includes information from 8 sensors, deployed on an expressway (I-94) section without ramps and traffic lights. The segment contains 7 road connections of 5.3 km at total (757 m in average) and the speed limit of 60 mph. The road segment and corresponding weighted directed graph are presented in Figure 1.

The sequential locations of road sensors were selected as a representative of the simplest spatial structure—there is only one possible path on the corresponding road graph.

The second segment road network segment represents the complex (typical) spatial structure and includes information from 19 sensors, deployed on expressways near to the city centre. Sensors serve different directions, so have different daily patterns. The segment contains 19 road connections of 30.9 km at total (1626 m in average) and speed limits of 55–60 mph. Four sensors (S138, S288, S584, S62) represent entrances to the analysed network segment, six sensors (S123, S283, S291, S586, S557, S566) represent exits, and another ten sensors are intermediate (have both incoming and outgoing links). The road segment and corresponding weighted directed graph are presented in Figure 2.

Traffic flows within the complex spatial structure are more diverse—the number of possible paths on the corresponding road graph is 11.

The purpose of conducting analysis for two road segments is related to existing trends in spatiotemporal traffic forecasting: many earlier studies are based on the sequential settings, while recently the focus has shifted to city-wide complex road networks [3]. We tested our research hypothesis in sequential and complex spatial settings to demonstrate the importance of temporal aggregation for both of them.

Daily patterns of traffic volume at different locations are naturally similar in the sequential spatial structure, but vary in the complex one. Figure 3 represents different daily patterns at selected sensors in the complex spatial structure: S93 (intermediate on the expressway) has relatively stable loading during the day, S291 (exit to the city centre) has a morning peak, and S123 (exit in a direction, opposite to the city centre) has a higher loading during the evening hours. 

We executed the research separately for original time series of traffic volume and detrended ones. Missing data were replaced by an average value of the 5-period window around the missed time point.

### 2.2. Data Preprocessing: Temporal Aggregation and Trend Removal

Let us define a traffic flow at a sensor location *i* as a counting process $N_{i}\left(t\right)$, representing the number of vehicles that passed through the sensor during the time interval (0, *t*). Then, time aggregation of this process with a level Δt (in minutes) is a time series, defined as:(1)$$y_{i,\;t}^{\Delta t}=N_{i}\left(t\Delta t\right)-N_{i}\left(\left(t-1\right)\Delta t\right)$$

In this research, we considered 6 levels of temporal aggregation, Δt∈{1, 2, 3, 4, 6, 12}. The selection of these values is explained by the research problem of short-term traffic forecasting (usually researchers consider intervals up to 15 min as a short term for traffic flows) and least common multiple of values (12 min).

Due to the additivity of the counting process, time series for different temporal aggregation levels Δt and KΔt are linked as:(2)$$y_{i,\;t}^{K\Delta t}=\sum_{k=0}^{K-1}y_{i,\;t-k}^{\Delta t}$$

Given a total observation time period T, the length of the aggregated time series is defined as $T^{\Delta t}=T\;div\;\Delta t$, where div is the integer division operator.

Generally, traffic flow is a multi-dimensional variable, which includes traffic volume, speed, headway, and other traffic characteristics. In this research, we concentrated on the analysis of one characteristic, traffic volume. There is a large potential of the modelling of traffic flow as a multivariate variable, and the generalization of all considered models for the multivariate case is straightforward.

The time series of traffic volumes contains natural daily, weekly, and yearly trends. There is empirical evidence showing that the removal of these trends can lead to better forecasting accuracy of models [53] and easier identification of spatial correlation between sensor locations [24]. We considered both original (not transformed) and detrended time series of the traffic flow for our analysis for the reduction in trend-related findings (it is well-known that trends in time series increase the risk of spurious regressions, but, on the other hand, trends can contain information that can be potentially useful for model forecasting performance). The analysed time interval is fairly short (5 weeks), and relates to a homogeneous period (March), so we removed trends on a weekly basis. Given the number of time points *v* per week as V= 7×24×60/Δt and the number of weeks as s=T div (7×24×60×Δt), we estimated trends as simple average values for a specified time interval of a week at a location *i*:(3)$$Trend_{i,v}^{\Delta t}=\;\frac{1}{s}\sum_{t=0}^{s-1}y_{i,v+tV}^{\Delta t}$$
and a detrended time series as
(4)$$y_{i,t}^{\Delta t,\;resid}=y_{i,t}^{\Delta t}-Trend_{i,t\;mod\;V}^{\Delta t}$$
where *mod* is the modulo operator. We utilized a separate set of 4 training weeks before the main sample for identifying weekly trends (using of the full sample for identification of weekly trends can lead to a bias for a short training period). For longer time series, it is recommended to use differencing with lag V or seasonal exponential smoothing [14] for detrending. All further research steps were separately applied for the original and detrended traffic volumes. 

Traffic patterns vary for working days and weekends. We excluded Sundays and Saturdays from the sample to reduce this source of variability.

### 2.3. Spatiotemporal Forecasting Models

The naïve forecasting model was used as a baseline for the model forecasting accuracy. We used basic (non-seasonal) naïve forecasts, where the last available value was used as a forecast for all forecasting horizons. The popular univariate Box–Jenkins method [54] was used as a univariate basis for forecasting accuracy comparison. The method includes specification and estimation of the autoregressive integrated moving average (ARIMA) model:(5)$$\Delta^{d}y_{t}=\sum_{h=1}^{p}\phi_{h}\Delta^{d}y_{t-h}+\varepsilon_{t}+\sum_{h=1}^{q}\theta_{h}\varepsilon_{t-h}$$
where *t* = 1, …,$\;T^{\Delta t}$; p and q are orders of autoregressive and moving average components respectively; $\Delta^{d}y_{t}$ is the dth order difference (stationary) of the time series $y_{t}$, integrated of order d;$\;\phi_{h}$ and $\theta_{h}$ are unknown model parameters; and $\varepsilon_{t}$ is an independent, identically normally distributed error with zero mean. The univariate model was separately applied for time series at every sensor location. An appropriate model specification was selected on the basis of Hyndman and Khandakar algorithm [55], which combines a repeated KPSS test [56] for identification of the time series order of integration and the corrected Akaike information criterion (AICc) for selection of autoregressive (p) and moving average (q) lag orders. 

A univariate ARIMA model can be considered as a good benchmark, which does not take spatial dependencies into account. An opposite approach is to allow all possible dependencies between time series, without utilizing information about the spatial structure. The vector autoregressive (VAR) model is widely used to capture the linear interdependencies among multiple time series. Let $Y_{t}$ be a k×1 vector $\left(y_{1,t},y_{2,t},\ldots ,y_{k,t}\right)’$, where $y_{i,t}$ is a value at a time point *t* and a spatial location *i*, then VAR(p) model is presented as:(6)$$Y_{t}=\mu +\sum_{h=1}^{p}\Phi_{h}Y_{t-h}+\varepsilon_{t},$$
where $\Phi_{h}$ is a set of k×k matrices of unknown coefficients for every lag h=1,…,p; μ is a k×1 vector of constants; $\varepsilon_{t}$ is a k×1 vector of i.i.d. errors with zero mean and a k×1 vector of variances $\sigma_{\varepsilon }^{2}$. For our research datasets, VAR models with the constant term do not outperform VAR models with μ=0, so further we refer to VAR(p) without the constant term as VAR for brevity reasons. A set of unknown parameters of VAR(p) model includes $\mu ,{\;\Phi }_{h}$, and $\sigma_{\varepsilon }^{2}$, and is usually fairly large ($k^{2}p+2k$ parameters). Due to the large number of parameters, VAR models suffer from the overfitting problem, which, in particular, affects their out-of-sample forecasting accuracy. Although several statistical regularization approaches (e.g., lasso [14]) can be used to reduce the number of estimated parameters, we concentrate on VAR models exogenously restricted by the spatial structure [20]. 

The primary object of this research is spatiotemporal models, the forecasting performance of which naturally depends on the spatial structure of the dataset. Many studies consider the physical properties of traffic flows and estimate spatial dependencies between consecutive points on arterial roads. However, the problem is more complex: different segments of a road network can be considered as complementary, and traffic flows could be relocated under specific conditions (congestion, traffic jams, etc.).

We conduct this research for two different spatial settings:Sensors located on an urban arterial road without ramps and traffic lights (the sequential structure)Sensors located on a typical urban road network (the complex structure)

Spatial dependencies in the sequential structure are highly expected due to the nature of traffic flow, and their existence is empirically supported by many previous studies. In its turn, spatial dependencies in the complex structure are a matter of uncertainty. In addition to the physical properties of traffic flows and related drivers’ decisions, these dependencies can appear between physically unconnected locations (teleconnections), explained by the simultaneous effects (i.e., dinner hours). Risks of spurious dependencies in the complex structure are higher, and the usefulness of the spatial structure for increasing the forecasting accuracy is not obvious and a matter of the selected temporal aggregation level.

Let $S_{h}=\left\{s_{h,\;ij}\right\}$ be a k×k binary matrix that represents possible spatial dependencies between values of the vector $Y_{t}$ and its lag of order *h*, $Y_{t-h}$ ($s_{h,ij}=1$ if a dependency between $y_{i,t}$ and $y_{j,t-h}$ is allowed and $s_{h,ij}=0$ if the dependency is impossible). Then the spatially restricted specification of VAR model is:(7)$$Y_{t}=\sum_{h=1}^{p}\left(\Phi_{h}\circ S_{h}\right)Y_{t-h}+\varepsilon_{t},$$
where $\Phi_{h}\circ S_{h}$ is the Hadamard (entry-wise) product of matrices. We refer to this model as spatially restricted VAR (SRVAR). Note that $S_{h}$ matrixes are usually sparse, so the number of unknown parameters is significantly smaller than in the unrestricted VAR model.

There are different approaches to specification of $S_{h}$ matrices. Min and Winter [19] used travel times between sensor locations (for upstream links), taking into account different traffic speed conditions. Yang et al. [21] and Cai et al. [57], among several other researchers, utilized cross-correlations for identification of “connected” road segments. Schimbinschi et al. [20] used direct road connections between sensor locations (both up- and downstream) for their topology-regularized VAR. We consider two alternative specifications of $S_{h}$ matrices: travel-time-based and correlation-based.

Travel-time-based approach. Let $d_{ij}$ be a distance by road between locations *i* and *j* (in meters), $speed_{ij}$ is an average speed at this road segment (m/min), and Δt is the temporal aggregation level (in minutes). Given that for the analysed road network the average speed is normally 10% higher than the speed limit and the empirical distribution of speed is left-tailed, we can define
(8)$$minLag_{ij}=\frac{d_{ij}}{\left(speed_{ij}\cdot \Delta t\right)},$$
as a minimal temporal lag between $y_{i}$ and $y_{j}$, allowed by the road network configuration (⌊…⌋ brackets are used for rounding down). Further, we set $s_{h,ij}$ to 1 if $minLag_{ij}=h$ for a travel-time-based specification of the matrix of spatial dependencies $S_{h,travel\;time}$. Using such specification, we assume that a spatial effect of time lag *h* between $y_{i}$ and $y_{j}$ is only possible if the point *i* is reachable from the point *j* in *h* time steps. The VAR model, spatially restricted by the specified travel-time matrix $S_{h,travel\;time}$, is referred to as ${\mathrm{SRVAR}}_{travel\;time}$.

Correlation-based approach. A concurrent SRVAR model specification is based on cross-correlations between time series at different locations. Let
(9)$$minLag_{ij}=\underset{h\in \left[-hlim,hlim\right]}{\mathrm{argmax}}Corr\left(y_{i,t},y_{j,t-h}\right),$$
where *hlim* is a predefined maximum possible lag and $Corr\left(y_{i,t},y_{j,t-h}\right)$ is a Pearson cross-correlation between time series. $minLag_{ij}$ is intentionally set to 0 if the maximum cross-correlation value is lower than a predefined threshold (we used 0.1 as a threshold in this research). Further specification of the restricted VAR model is identical to ${\mathrm{SRVAR}}_{travel\;time}$; the resulting VAR model is referred to as ${\mathrm{SRVAR}}_{correlation}$.

The presented spatially restricted VAR model should not be confounded with spatial VAR models, SVAR, or with space–time autoregressive models, STAR. SVAR [58] contains a simultaneous (not temporarily lagged) spatially lagged dependent variable, which makes them less natural for forecasting purposes than the presented SRVAR model. STAR models [9] represent a restricted form of the SRVAR model, where the $\Phi_{h}\circ S_{h}$ matrix is reduced to $\phi_{h}W_{h}$ ($W_{h}$ is an exogenous matrix of spatial weights). STAR models are more parsimonious in terms of unknown parameters, but require careful specification of spatial weights, which can be a problem for the dynamic environment.

Summarising the methodology above, the final set of candidate models is:Naïve forecasts as a baseline,Independent ARIMA models are a good non-spatial alternative,Travel-time-based spatially restricted VAR model, ${\mathrm{SRVAR}}_{travel\;time}$, Correlation-based spatially restricted VAR model, ${\mathrm{SRVAR}}_{correlation}$,Unrestricted VAR model.

### 2.4. Estimation of Forecasting Accuracy

Overfitting is a well-known problem for multivariate models, including VAR, so in-sample metrics of accuracy should not be used for model comparison. We applied the rolling analysis [59], a time series cross-validation technique, for the estimation of models’ out-of-sample forecasting accuracy. A rolling window is a time subinterval of a predefined length $T_{RW}$ (the number of points in the corresponding aggregated time series is $T_{RW}^{\Delta t}=T_{RW}\;div\;\Delta t$), which is used as a training set for models. The rolling window is gradually shifted inside the research time interval of a step Δt, providing new model estimates and forecasts. We use the same forecasting horizon *h* for models of all temporal aggregation levels, so the total length of the validation set (and the number of the rolling window shifts) equals $T_{V}^{\Delta t}=T^{\Delta t}-T_{RW}^{\Delta t}-h+1$.

The mean absolute error (MAE) is used in this research as an absolute metric of the model forecasting accuracy:(10)$$MAE\left(i,h,\Delta t\right)=\frac{1}{T_{V}^{\Delta t}}\sum_{t=T_{RW}^{\Delta t}}^{T^{\Delta t}-h}\left|y_{i,\;t+h}^{\Delta t}-{\hat{y}}_{i,\;t+h|t}^{\Delta t}\right|,$$
where ${\hat{y}}_{i,\;t+h|t}^{\Delta t}$ are *h*-step-ahead predicted values. Furthermore, MAE(i,h,Δt) are aggregated by spatial locations with equal weights: $MAE\left(h,\Delta t\right)=\frac{1}{k}\sum_{i=1}^{k}MAE\left(i,h,\Delta t\right)$. A widely used relative metric for forecasting accuracy comparison is the mean absolute percentage error (MAPE). Despite the popularity of this metric, there are many studies on misleading results from MAPE, obtained for time series with close-to-zero (or zero) actual values. As the traffic volume (both original and detrended) is naturally allowed to be 0, we prefer the mean absolute scaled error (MASE) [60] as the main metrics for comparison of different models:(11)$$MASE\left(i,h,\Delta t\right)=\frac{MAE\left(h\right)}{\frac{1}{T_{V}^{\Delta t}-1}\sum_{t=T_{RW}^{\Delta t}+1}^{T^{\Delta t}-h}\left|y_{i,\;t+h}^{\Delta t}-y_{i,\;t+h-1}^{\Delta t}\right|},$$

MASE is a relatively new forecasting accuracy metric, which is based on the comparison of the model’s MAE values with one-step naïve forecasting MAE values. This metric has good statistical properties and allows direct comparison of forecasting models. Similar to MAE, MASE values are separately calculated for every sensor location and aggregated for the overall model forecasting accuracy measure.

To compare the forecasting accuracy of models, estimated for different time aggregation levels, we used the sums of forecasts for shorter time frames ${\hat{y}}_{i,\;t}^{K\Delta t}=\sum_{k=0}^{K-1}{\hat{y}}_{i,\;t-k}^{\Delta t}$ and the corresponding cumulative MAE metric. For example, to estimate the forecasting accuracy of a model, estimated for 4 min temporal aggregation, for a 12 min horizon, we applied MAE(i,h,3Δt) values. 

## 3. Results and Discussion

The primary research question relates to the forecasting accuracy of spatiotemporal models for different temporal aggregation levels. We consider three different spatiotemporal specifications of VAR models: the travel-time-based and correlation-based spatially restricted VAR and the unrestricted VAR model. Principles of both approaches to spatial restriction are described in the Methodology section.

Both travel-time- and correlation-based restrictive matrices depend on the temporal aggregation level. Table 1 contains estimated matrices for original traffic volumes in the sequential spatial settings for 1-min temporal aggregation (and a matrix of most significant lags in the unrestricted VAR model for comparison). 

It can be easily noted that travel-time- and correlation-based restrictive matrixes are almost identical in this case. The unrestricted VAR specification discovers a lot of additional spatial dependencies, which cannot be explained by the kinematic traffic flow theory.

In the complex spatial settings, these matrices differ significantly—Table 2 presents the number of links in matrices for different temporal aggregation levels and dissimilarity (percent of different links) between travel-time- and correlation-based matrixes. 

In the complex spatial environment, travel-time restriction is the strictest one—for example, for the 12 min temporal aggregation, the travel-time restrictive matrix does not allow any dependencies between locations, because all vehicles should leave the analysed road segment in 12 min. Complete results for all temporal aggregation levels and both spatial environments are available from the author by request.

MASE values of five models (naïve forecasts, univariate ARIMA, travel-time-based SRVAR, correlation-based SRVAR, and unrestricted VAR) were estimated for different temporal aggregation levels, using the rolling window technique, described in the Methodology section. The accuracy of one-step ahead forecasts of detrended traffic volumes is plotted in Figure 4; complete calculation results for four different experiment settings (original and detrended traffic volumes for the sequential and complex spatial structure) are presented in Table 3.

The first notable observation is a better forecasting performance of spatiotemporal models (comparatively to the independent ARIMA models) for smaller periods of temporal aggregation (1and 2 min periods). The unrestricted VAR model demonstrates the best forecasting performance for the sequential spatial structure, while for the complex spatial structure, it does not outperform SRVAR model specifications. For larger temporal aggregation levels, the forecasting accuracy of SRVAR and independent ARIMA models are very similar, while the accuracy of the unrestricted VAR model is significantly degraded. In addition, we note that this pattern (preference of spatiotemporal models for smaller temporal aggregation levels) is stable within this research, but obviously depends on the spatial environments. The average travel time between sensor locations in the complex spatial structure is about 1 min, and the maximum is about 6 min, which corresponds to temporal aggregation levels with discovered preference of spatiotemporal models.

In addition, it should be noted that for larger temporal aggregation levels, ${\mathrm{SRVAR}}_{travel\;time}$ utilizes a very limited number of allowed lags (eight lags for 6 min periods, and no lags for 12 min periods), so we can conclude that in terms of the forecasting accuracy, information about the absence of spatial dependencies is equally or even more important than information about their presence.

Another observation is almost identical forecasting accuracies of correlation-based (${\mathrm{SRVAR}}_{travel\;time}$) and travel-time-based (${\mathrm{SRVAR}}_{travel\;time}$) spatially restricted VAR models. This observation is essential for the sequential spatial settings, where correlation-based and travel-time-based spatial restrictive matrixes are almost identical, but was not expected in the complex spatial settings, where the matrices differ significantly. Both approaches to spatial restriction have their own advantages (flexibility and endogeneity of correlation-based and manageability of the travel-time-based restrictions), so we can conclude that this choice is experiment-specific and depends on available data and modelling purposes.

The dynamics of the models’ forecasting accuracy for different horizons are presented in Figure 5. 

The results are consistent between the sequential and complex spatial settings: better forecasting performance of spatiotemporal models for 1 and 2 min forecasting horizons, and a small preference of independent ARIMA models for longer horizons (due to their higher flexibility). 

The obtained results represent the forecasting accuracy, averaged for analysed sensors. Obviously, improvement of spatial model specifications depends on sensor position within the network. For example, ${\mathrm{SRVAR}}_{travel\;time}$ includes only the forward propagation of traffic volume conditions, and thus it cannot improve forecasting accuracy for sensors, located at entrances to the analysed network segment. Table 4 contains sensor-specific forecasting accuracy values for one-step ahead forecasts of the models with 1 min temporal aggregation. 

The results for the sequential spatial structure directly match our expectations: forecasting accuracy for first two sensors in a row (S240 and S242) has similar values for independent ARIMA and SRVAR models, but for other sensors the improvement, provided by spatiotemporal specifications, is significant (up to −0.27 of MASE for the detrended and up to −0.41 for the original traffic volume). Results for the complex spatial structure are also consistent—no improvements for the entrance sensors (S138, S288, S584, S62) and significant improvement for the majority of other sensors. The most interesting are the exceptions—no significant improvements are discovered for intermediate S93 and S103 sensors. Both sensors are located in I-94 E/WB corridor, which crosses the analysed road segment at the western part. The absence of improvement in spatiotemporal models for these sensors means that traffic flow in this corridor mostly depends on other roads, which pass outside of the analysed road segment. Thus, observed forecasting accuracy of spatially restricted VAR models allows to us identify anomalies in the spatial structure and to provide a potential method for model improvement.

Finally, we analysed the forecasting accuracy of the models for a longer period of time on the basis of different temporal aggregation levels. We selected 12-min interval for comparison as the least common multiple of research temporal aggregation levels (1, 2, 3, 4, 6, and 12 min). Forecasts for smaller temporal aggregation levels were aggregated as described in the Methodology section (by direct summation of the necessary number of multi-step-ahead forecasts). Dependencies between resulting forecasting accuracy and a temporal aggregation level are presented in Figure 6.

Presented polylines provide evidence for a statistical preference of 2, 3, 4, and 6 temporal aggregation levels for 12 min forecasting with a model-dependent optimal time aggregation level. For example, the unrestricted VAR model demonstrates the best forecasting accuracy for the 4 min temporal aggregation level. These results match our expectations, based on previous studies of univariate and multivariate time series models [61]. It should be stressed that our results are empirical and cannot be proven theoretically, because models for different temporal aggregation levels are not identical and depend on the aggregation-specific spatial restrictions. Nevertheless, the provided evidence of the existence of an optimal temporal aggregation level could be useful for practical forecasting. Further improvements include the application of a multi aggregation prediction algorithm, described in [48], for combining forecasts from models, estimated for different temporal aggregation levels.

The results for original time series are generally consistent with the findings presented above. It should be noted that for the majority of executed experiments, the forecasting accuracy of the detrended traffic volume is higher than of the original time series. This result is consistent with the recommendations provided in [24,53]. Additionally, we tested the research hypothesis separately for the peak hours, which are most important in terms of traffic forecasting. The general conclusions were very close to the complete dataset. This fact was expected, because we focus on average forecasting error metrics (MAE, MASE) and all selected models (including the naïve forecasts) work well during “calm” periods, so the major part of the average error appears during peak hours or unusual traffic conditions. Given similar results and almost identical conclusions for a complete dataset and for peak hours, we decided to limit the paper text and to leave only complete dataset results.

Summarising the results and discussion above, we state the key empirical findings as follows:Unrestricted VAR models outperform spatially restricted models in the sequential spatial structure with strong essential spatial links for smaller temporal resolution levels. In the complex spatial structure, spatially restricted models are as good as unrestricted VAR models for smaller temporal resolution levels and significantly outperform them for larger temporal resolution levels (with a limited number of possible spatial links). Thus, we strongly recommend spatial restrictions for the modelling of traffic flows in a highly connected urban spatial environment.Correlation-based spatial restrictive matrixes are almost identical to travel-time-based ones for the sequential spatial structure. For the complex spatial structure, travel-time-based spatial restrictive matrices are significantly sparser, but this fact does not negatively affect model out-of-sample forecasting accuracy. Thus, we recommend the travel-time-based spatial restrictions as a baseline for spatiotemporal model specifications.Improvements of sensor-specific forecasting accuracy, provided by the spatiotemporal model specifications, depend on a sensor position within the road network graph. These improvements are highly significant for sensors, located inside an analysed network segment and on its exits, and insignificant for sensors, located near entrances to the analysed network segment. We concluded that insignificant improvements for sensors, located inside the analysed network segment, could be considered as an indicator of an incomplete spatial structure.Optimal temporal aggregation levels for longer forecasting horizons (12 min in this research) are found at intermediate values (2–6 min) and related to the spatial structure (average distances between spatial locations). We recommend considering different temporal aggregation levels for any modelling approaches, and concentrate on temporal aggregation levels, which are close to spatial resolution (travel times between spatial locations), for spatiotemporal models.The majority of executed experiments indicate higher forecasting accuracy for models of detrended traffic volumes comparatively to models of the original time series.

## 4. Conclusions

This study is devoted to the empirical analysis of temporal aggregation effects of forecasting accuracy of spatiotemporal models. We demonstrated that a selected level of temporal aggregation directly affects the structure of spatiotemporal relationships and, as a result, the forecasting performance of models. The effects of temporal aggregation vary for different model specifications and different forecasting horizons. Thus, we argue that the level of temporal aggregation should be included in the primary set of tuned parameters, used for the optimisation of the model’s forecasting performance and selection of the best model. Analysis of models’ forecasting accuracy for different temporal aggregation levels improves the confidence in the obtained results and allows wider generalisation of the conclusions made.

We empirically tested unrestricted and spatially restricted vector autoregressive models and compared their forecasting accuracy for different temporal aggregation levels of real-world traffic flow data. The research methodology is based on the rolling window analysis of out-of-sample model forecasting performance. Research experiments were executed in several dimensions: temporal aggregation levels (1, 2, 3, 4, 6, and 12 min), forecasting horizons (one-step and multi-step forecasts), spatial complexity (sequential and complex spatial structures), spatial restriction approach (unrestricted, travel-time-based and correlation-based), time series transformation (original and detrended traffic volumes). The empirical results support our main proposition on the crucial role of the temporal aggregation level.

Although the empirical part of this study is purely focused on urban traffic data analysis, the discussed problem is applicable to other multi-sensor systems, where relationships between sensor data streams play an important role, e.g., medical wearable sensors or systems of meteorological stations. In addition to the forecasting model selection problem discussed in this paper, the selection of the optimal temporal aggregation level can be important for reducing redundant information transmissions and limiting network traffic flows.

This study analyses the temporal aspect of data aggregation only, while aggregation in the spatial dimension (e.g., spatial clustering of sensor data) also plays the important role for modern spatiotemporal traffic forecasting models. A detailed analysis of the spatiotemporal aggregation effects is required and can be mentioned as a direction for further research.

## Figures and Tables

**Figure 1 sensors-20-06931-f001:**
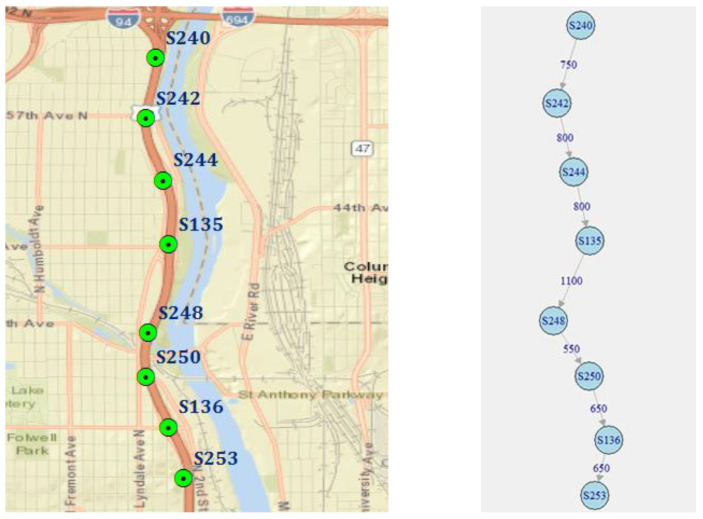
Road segment with sequential sensor locations: a map and corresponding weighted directed graph (weights represent distances by road in meters).

**Figure 2 sensors-20-06931-f002:**
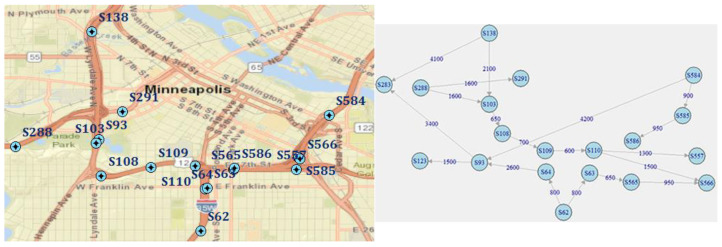
Road segment with non-sequential sensor locations: a map and corresponding weighted directed graph (weights represent distance by road in meters).

**Figure 3 sensors-20-06931-f003:**
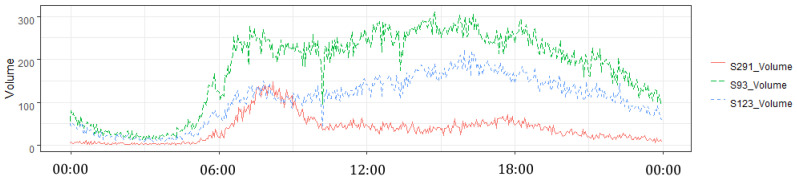
Typical daily traffic at three sensor locations (S291, S93, and S123).

**Figure 4 sensors-20-06931-f004:**
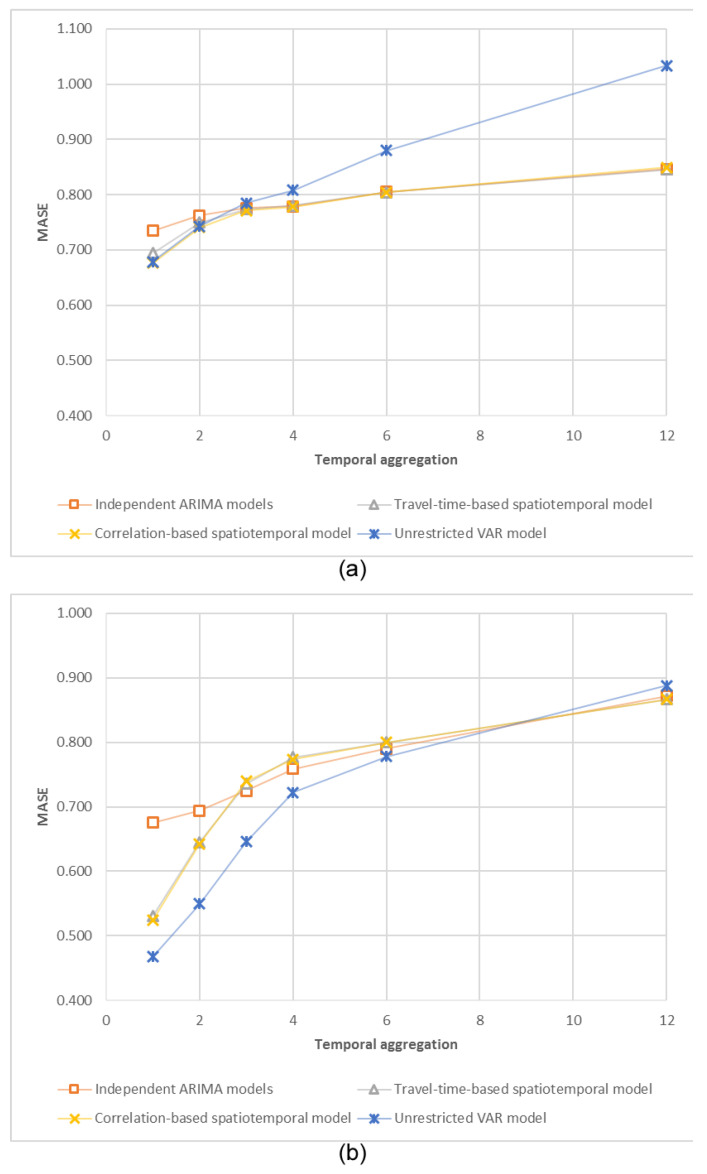
MASE values for one-step ahead forecasts for different temporal aggregation levels: (**a**) detrended traffic volume, the complex spatial structure, (**b**) detrended traffic volume, the sequential spatial structure.

**Figure 5 sensors-20-06931-f005:**
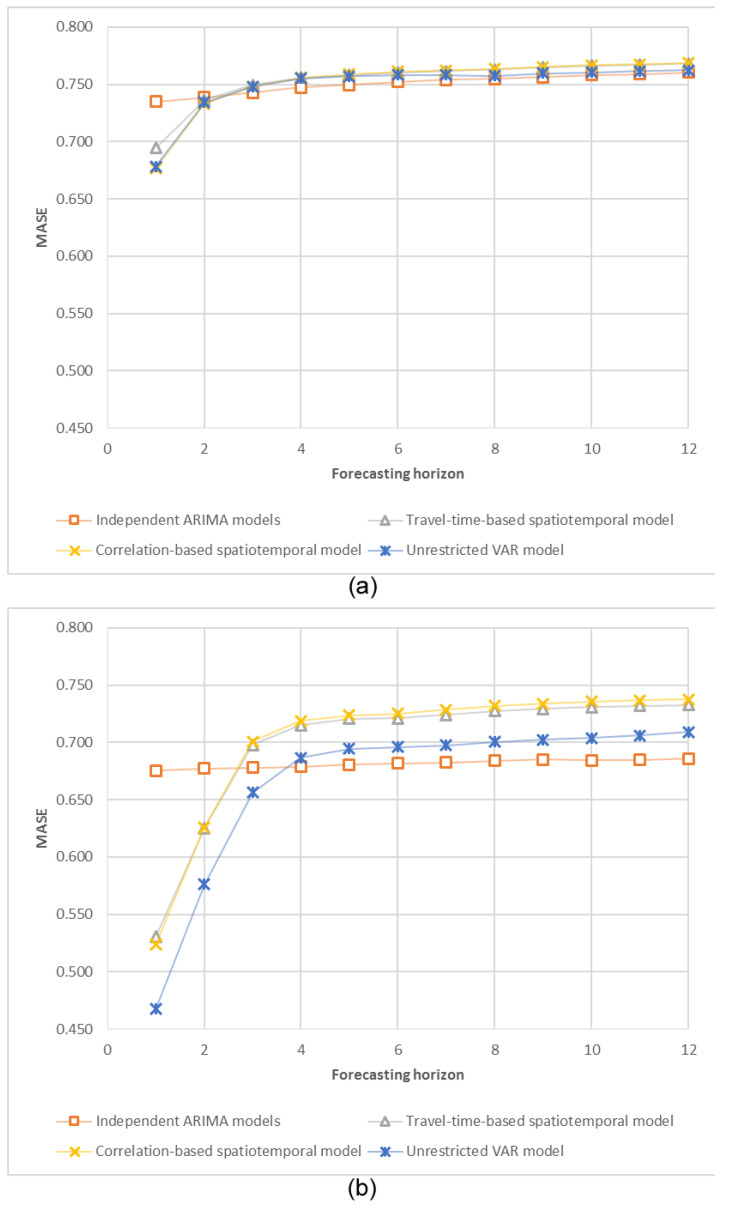
MASE values for different forecasting horizons: (**a**) detrended traffic volume, the complex spatial structure, (**b**) detrended traffic volume, the sequential spatial structure.

**Figure 6 sensors-20-06931-f006:**
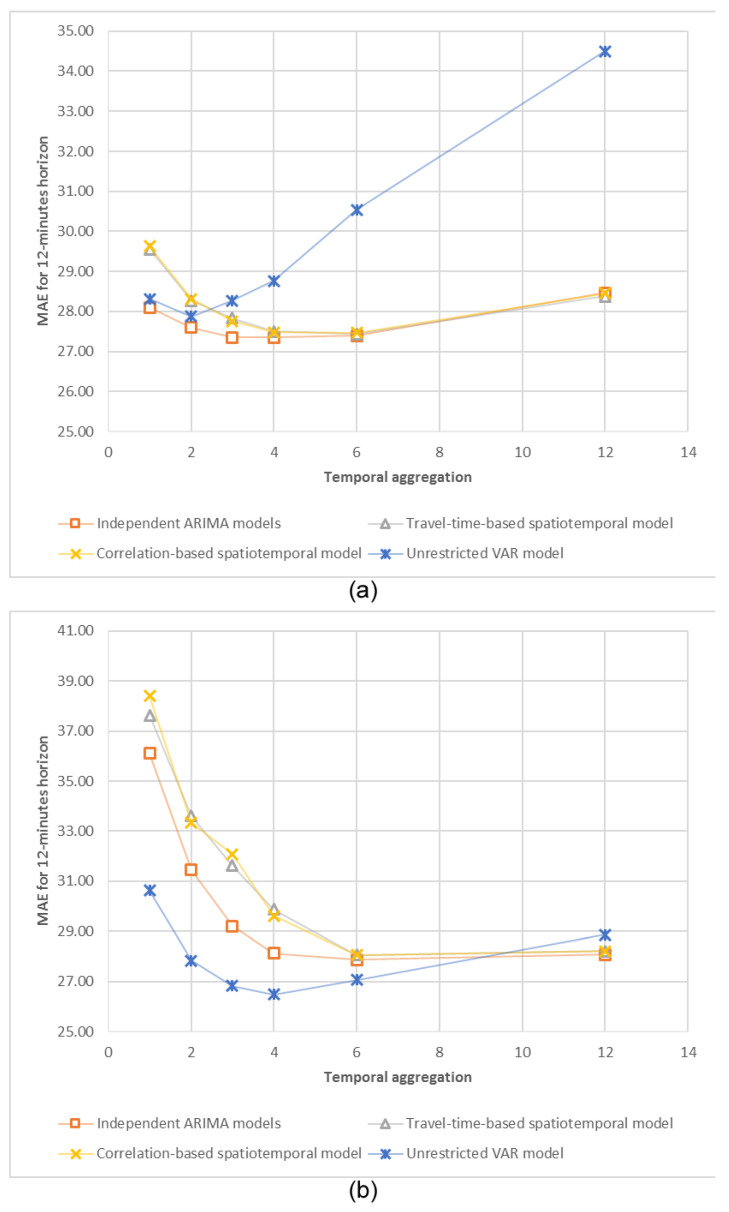
MAE values for 12 min forecasts for different temporal aggregation levels: (**a**) detrended traffic volume, the complex spatial structure, (**b**) detrended traffic volume, the sequential spatial structure.

**Table 1 sensors-20-06931-t001:** Matrices of spatial lags for the original traffic volume in the sequential spatial structure.

Travel-Time-Based Spatial Restrictions								
	S240	S242	S244	S135	S248	S250	S136	S253
S240	0	0	1	1	2	2	3	3
S242	0	0	0	1	2	2	2	3
S244	0	0	0	0	1	1	2	2
S135	0	0	0	0	1	1	1	2
S248	0	0	0	0	0	0	1	1
S250	0	0	0	0	0	0	0	1
S136	0	0	0	0	0	0	0	0
S253	0	0	0	0	0	0	0	0
**Correlation-Based Spatial Restrictions**								
	S240	S242	S244	S135	S248	S250	S136	S253
S240	0	0	1	1	2	2	3	3
S242	0	0	0	1	2	2	2	3
S244	0	0	0	1	1	1	2	3
S135	0	0	0	0	1	1	1	2
S248	0	0	0	0	0	0	1	1
S250	0	0	0	0	0	0	1	1
S136	0	0	0	0	0	0	0	1
S253	0	0	0	0	0	0	0	0
**Unrestricted VAR, Most Significant Lags (std. Error > 3)**								
	S240	S242	S244	S135	S248	S250	S136	S253
S240	4	1	1	1	2	2	3	2
S242	1	1	1	1	2	2	2	3
S244	3	6	1	1	1	1	2	3
S135	0	2	1	1	1	1	1	2
S248	0	0	0	1	4	1	1	4
S250	6	6	4	1	2	1	1	1
S136	1	6	6	3	0	1	1	1
S253	3	0	0	6	6	6	0	4

**Table 2 sensors-20-06931-t002:** Number of links and similarity of spatial dependency matrices.

Temporal Aggregation, Minutes	Unrestricted VAR	Correlation-Based SRVAR	Travel-Time-Based SRVAR	Percent of Different Links in Correlation-Based and Travel-Time-Based Matrices
**The complex spatial structure, detrended traffic flow**				
1	96	31	43	6.6
2	88	24	31	5.8
3	85	26	24	7.8
4	63	38	18	12.7
6	67	48	8	15.5
12	52	71	0	19.7
**The complex spatial structure, original traffic flow**				
1	122	145	43	29.4
2	115	129	31	28.8
3	112	103	24	23.0
4	111	91	18	21.3
6	89	76	8	19.4
12	86	62	0	17.2
**The sequential spatial structure, detrended traffic flow**				
1	52	25	22	4.7
2	46	17	16	1.6
3	35	13	11	3.1
4	30	7	6	1.6
6	22	1	0	1.6
12	13	0	0	0.0
**The sequential spatial structure, original traffic flow**				
1	56	25	22	4.7
2	50	17	16	1.6
3	43	12	11	1.6
4	39	5	6	1.6
6	27	0	0	0.0
12	21	0	0	0.0

**Table 3 sensors-20-06931-t003:** Mean absolute scaled error (MASE) values for one-step ahead forecasts for different temporal aggregation levels.

	Temporal Aggregation, Minutes					
	1	2	3	4	6	12
**The complex spatial structure, detrended traffic flow**						
Naïve forecasts	1.000	1.000	1.000	1.000	1.000	1.000
Independent ARIMA models	0.735	0.762	0.775	0.779	0.805	0.846
Travel-time-based SRVAR model	0.694	0.750	0.774	0.780	0.804	0.846
Correlation-based SRVAR model	0.676	0.741	0.771	0.778	0.804	0.849
Unrestricted VAR model	0.678	0.743	0.785	0.808	0.880	1.033
**The complex spatial structure, original traffic flow**						
Naïve forecasts	1.000	1.000	1.000	1.000	1.000	1.000
Independent ARIMA models	0.776	0.848	0.903	0.944	0.960	0.999
Travel-time-based SRVAR model	0.730	0.798	0.850	0.887	0.947	0.967
Correlation-based SRVAR model	0.701	0.781	0.837	0.869	0.915	0.928
Unrestricted VAR model	0.691	0.766	0.828	0.875	0.936	1.015
**The sequential spatial structure, detrended traffic flow**						
Naïve forecasts	1.000	1.000	1.000	1.000	1.000	1.000
Independent ARIMA models	0.675	0.694	0.725	0.758	0.790	0.872
Travel-time-based SRVAR model	0.530	0.645	0.736	0.777	0.800	0.867
Correlation-based SRVAR model	0.524	0.642	0.740	0.773	0.800	0.867
Unrestricted VAR model	0.468	0.550	0.646	0.722	0.778	0.888
**The sequential spatial structure, original traffic flow**						
Naïve forecasts	1.000	1.000	1.000	1.000	1.000	1.000
Independent ARIMA models	0.697	0.691	0.871	0.992	0.927	0.954
Travel-time-based SRVAR model	0.437	0.500	0.756	0.814	0.925	0.907
Correlation-based SRVAR model	0.421	0.500	0.753	0.857	0.936	0.907
Unrestricted VAR model	0.397	0.472	0.670	0.754	0.867	0.953

**Table 4 sensors-20-06931-t004:** Sensor-specific MASE values.

	Detrended Traffic Volume						Original Traffic Volume					
	Naive Forecasts	ARIMA	Travel-Time-Based SRVAR Model	Correlation-Based SRVAR Model	Unrestricted VAR Model	SRVAR vs. ARIMA	Naive Forecasts	ARIMA	Travel-Time-Based SRVAR Model	Correlation-Based SRVAR Model	Unrestricted VAR Model	SRVAR vs. ARIMA
**The complex spatial structure**												
S62	1.00	0.76	0.76	0.76	0.75	0.00	1.00	0.79	0.80	0.77	0.78	−0.02
S64	1.00	0.69	0.70	0.70	0.71	0.00	1.00	0.77	0.76	0.75	0.74	−0.02
S63	1.00	0.76	0.76	0.67	0.66	−0.09	1.00	0.80	0.81	0.69	0.67	−0.11
S93	1.00	0.74	0.74	0.73	0.74	0.00	1.00	0.78	0.78	0.77	0.76	−0.01
S103	1.00	0.77	0.76	0.76	0.74	0.00	1.00	0.80	0.76	0.76	0.73	−0.04
S109	1.00	0.72	0.66	0.64	0.65	−0.08	1.00	0.74	0.68	0.65	0.65	−0.09
S110	1.00	0.73	0.59	0.60	0.61	−0.13	1.00	0.75	0.63	0.62	0.62	−0.13
S123	1.00	0.70	0.62	0.62	0.62	−0.08	1.00	0.72	0.65	0.65	0.63	−0.07
S138	1.00	0.73	0.74	0.74	0.75	0.02	1.00	0.74	0.75	0.75	0.76	0.01
S283	1.00	0.72	0.72	0.71	0.70	−0.02	1.00	0.77	0.76	0.75	0.73	−0.01
S288	1.00	0.73	0.74	0.74	0.75	0.02	1.00	0.78	0.77	0.76	0.75	−0.02
S291	1.00	0.70	0.62	0.62	0.63	−0.07	1.00	0.74	0.71	0.67	0.65	−0.07
S557	1.00	0.70	0.56	0.56	0.57	−0.14	1.00	0.76	0.58	0.58	0.57	−0.18
S565	1.00	0.74	0.63	0.55	0.55	−0.19	1.00	0.79	0.68	0.58	0.56	−0.21
S566	1.00	0.72	0.62	0.62	0.62	−0.10	1.00	0.78	0.65	0.64	0.63	−0.14
S584	1.00	0.73	0.74	0.74	0.74	0.00	1.00	0.76	0.76	0.75	0.76	−0.01
S108	1.00	0.73	0.72	0.58	0.58	−0.15	1.00	0.77	0.73	0.58	0.58	−0.20
S585	1.00	0.75	0.71	0.71	0.71	−0.05	1.00	0.80	0.75	0.75	0.73	−0.05
S586	1.00	0.89	0.81	0.81	0.80	−0.08	1.00	0.89	0.84	0.83	0.81	−0.06
**The sequential spatial structure**												
S240	1.00	0.67	0.70	0.70	0.70	0.03	1.00	0.66	0.61	0.61	0.58	−0.06
S242	1.00	0.68	0.76	0.76	0.65	0.09	1.00	0.73	0.68	0.68	0.75	−0.05
S244	1.00	0.66	0.45	0.45	0.37	−0.21	1.00	0.67	0.28	0.28	0.26	−0.40
S135	1.00	0.68	0.56	0.55	0.43	−0.13	1.00	0.68	0.52	0.42	0.31	−0.26
S248	1.00	0.67	0.42	0.42	0.36	−0.24	1.00	0.70	0.29	0.29	0.29	−0.41
S250	1.00	0.67	0.42	0.40	0.39	−0.27	1.00	0.70	0.31	0.30	0.29	−0.40
S136	1.00	0.69	0.41	0.41	0.40	−0.27	1.00	0.72	0.33	0.34	0.33	−0.38
S253	1.00	0.69	0.52	0.48	0.44	−0.21	1.00	0.70	0.47	0.46	0.38	−0.24

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
