# Peer review of "Temporal Aggregation Effects in Spatiotemporal Traffic Modelling"

_sensors, 2020, doi:10.3390/s20236931_

Round 1
Reviewer 1 Report
The article is an empirical study of temporal aggregation effects in spatiotemporal modelling. The approach is interesting. To improve the paper I would suggest:
1- Improve the "Abstract": It should contain a summary of the background information, methods, results and conclusions. However, results and conclusions are approached only in the last sentence and very vaguely.
2- Show better in the text how the article differs from others already published and brings a significant contribution.
3- Different citation styles are used in the text. It is recommended to use the references in square brackets, numbered sequentially as they appear in the text. So, for example, on page 2 line 58 the citation Okutani and Stephanedes (1984) should be changed. The entire text must be revised to standardize all citations and place them in numerical sequence. For example, after citation [4], then we find citation[8] when it should be [5].
4- The journal’s recommendation is "All Figures, Schemes and Tables should be inserted into the main text close to their first citation". An Appendix was used to gather all the tables. An Appendix is ​​intended for additional, supplementary data, not essential to understanding and, possibly, that does not interest all readers. On page 9, line 355, the text "all tables are provided in Appendix", when, at that time, only table 1 had been mentioned, makes it appear that the Appendix had the purpose of only gathering the tables. The author must evaluate this situation and make any adjustments that are necessary for a better understanding of the reader.
5- A general revision of the text would be useful to improve the language, correct grammatical and typographical errors. Some examples for evaluation:
- a) On page 2 lines 52 to 56: Enter the Section number. All are "Section 0"; b) On page 2, line 81: "instable" - "unstable"; c) Page 3, line 111: "looks as" - "looks like"; d) Page 3, line 130: "advice" - "advise"; e) Page 8, line 317: "sensors that deployed in Minneapolis" - "sensors deployed in Minneapolis"; f) Page 12, line 448: "statistically preference" - "statistical preference"; g) Page 12, line 454: "depends of" - "depends on"
Author Response
Dear Reviewer,
Thank you for your valuable suggestions – please find my response and descriptions of changes made in the file attached

Reviewer 2 Report
This article presents a study devoted to empirical analysis of temporal aggregation effects of forecasting accuracy of spatiotemporal models. It is a somewhat confusing manuscript for the reader to understand because, there is variation in the proposal by the author and the reader will not understand what the purpose of the manuscript is. However, I will comment on certain errors or certain corrections to improve the quality of this article:
-The Abstract section is very weak because it does not communicate what the main issue is and what the author proposes to solve the problem.
-In the presentation of the Sections of this article, the author has forgotten to indicate the section number, since only 0 is placed. This paragraph should be included in the final part of Section 2. Related Works.
-The Introduction Section should be improved.
-In Section 2, the related works are misquoted.
-The titles of the Sections should be improved, they do not imply anything.
-The acronyms are misspelled, because the correct form is to write them with the first capital letter of the meaning of the acronym, as in line 318. Also, there are acronyms that do not coincide with their meaning as in line 90.
-What is the reason why the author chose the traffic volume data for those specific dates? Why was a volume data not chosen where there were peak hours?
-What is particular about the two routes analyzed for this study? Can it be replicated in other scenarios? What is necessary to replicate the same study in other scenario?
-What are the sensors that are deployed on the roads that are being analyzed?
-Was this study performed with any traffic or mobility simulator to validate what was studied?
-The tables that are in the appendix must be included in the manuscript, and explained by the author, answering that they wanted to do? What was obtained? If there is the possibility to create Figures that demonstrate and corroborate what is explained in those tables.
-Figures 1 to 5, should improve the size because the understanding is difficult and the legends, the axis titles are not understood.
-The conclusions must be improved, which are objective and clear to solve the problem posed by the manuscript. Also, there is no future work written in the Conclusions Section.
Author Response

(The authors gave the same response as above.)

Round 2
Reviewer 2 Report
Thanks to the authors for performing the changes suggested by the reviewers. Still pending is spell check.